# Genomic mutation profile in progressive chronic lymphocytic leukemia patients prior to first-line chemoimmunotherapy with FCR and rituximab maintenance (REM)

Julia González-Rincón[1,2], José A. Garcia-Vela[3], Sagrario Gómez[1], Belén Fernández-Cuevas[4], Sara Nova-Gurumeta[4], Nuria Pérez-Sanz[4], Miguel Alcoceba[2,5], Marcos González[2,5], Eduardo Anguita[6], Javier López-Jiménez[7], Eva González-Barca[8], Lucrecia Yáñez[9], Ernesto Pérez-Persona[10], Javier de la Serna[11], Miguel Fernández-Zarzoso[12], Guillermo Deben[13], Francisco J. Peñalver[14], María C. Fernández[15], Jaime Pérez de Oteyza[16], M. Ángeles Andreu[17], M. Ángeles Ruíz-Guinaldo[18], Raquel Paz-Arias[19], M. Dolores García-Malo[20], Valle Recasens[21], Rosa Collado[22], Raúl Córdoba[23], Belén Navarro-Matilla[4], Margarita Sánchez-Beato[1,2‡]*, José A. García-Marco[4‡]*

1 Lymphoma Research Group, Instituto de Investigación Sanitaria Puerta de Hierro-Segovia de Arana (IDPHISA), Madrid, Spain, 2 Centro de Investigación Biomédica en Red de Cáncer (CIBERONC), Madrid, Spain, 3 Hematology Department, Hospital Universitario de Getafe, Madrid, Spain, 4 Hematology Department, Hospital Universitario Puerta de Hierro-Majadahonda and Instituto de Investigación Sanitaria Puerta de Hierro-Segovia de Arana (IDIPHISA), Madrid, Spain, 5 Hematology, Hospital Universitario de Salamanca, Salamanca, Spain, 6 Hematology, Hospital Clínico San Carlos, IdISSC, UCM, Madrid, Spain, 7 Hematology, Hospital Ramon y Cajal, Madrid, Spain, 8 Hematology, Institut Català d'Oncologia, IDIBELL, L'Hospitalet de LLobregat, Spain, 9 Hematology, Hospital Universitario Marqués de Valdecilla, Santander, Spain, 10 Hematology, Hospital Txagorritxu, Vitoria, Spain, 11 Hematology, Hospital Doce de Octubre, Madrid, Spain, 12 Hematology, Hospital Universitario Dr Peset, Valencia, Spain, 13 Hematology, CHU Juan Canalejo, A Coruña, Spain, 14 Hematology, Fundación Hospital de Alcorcón, Madrid, Spain, 15 Hematology, Hospital Universitario Puerta del Mar, Cadiz, Spain, 16 Hematology, Hospital Madrid Norte Sanchinarro, Madrid, Spain, 17 Hematology, Hospital General de Móstoles, Madrid, Spain, 18 Hematology, Hospital Francesc Borja, Valencia, Spain, 19 Hematology, Hospital Universitario La Paz, Madrid, Spain, 20 Hematology, Hospital Morales Meseguer, Murcia, Spain, 21 Hematology, Hospital Miguel Servet, Zaragoza, Spain, 22 Citogenetics and molecular biology laboratory, Consorcio Hospital General Universitario, Valencia, Spain, 23 Hematology, Hospital Fundación Jiménez Díaz, Madrid, Spain

‡ MSB and JAGM contributed equally to this work as co-first authors.
* msbeato@idiphim.org (MSB); jagarciam@aehh.org (JAGM)

**Data Availability Statement:** Data have been deposited in the Sequence Read Archive (SRA)

## Abstract

Chronic Lymphocytic Leukemia (CLL) is the most prevalent leukemia in Western countries and is notable for its variable clinical course. This variability is partly reflected by the mutational status of IGHV genes. Many CLL samples have been studied in recent years by next-generation sequencing. These studies have identified recurrent somatic mutations in *NOTCH1*, *SF3B1*, *ATM*, *TP53*, *BIRC3* and others genes that play roles in cell cycle, DNA repair, RNA metabolism and splicing. In this study, we have taken a deep-targeted massive sequencing approach to analyze the impact of mutations in the most frequently mutated genes in patients with CLL enrolled in the REM (*rituximab en mantenimiento*) clinical trial. The mutational status of our patients with CLL, except for the *TP53* gene, does not seem to affect the good results obtained with maintenance therapy with rituximab after front-line FCR treatment.

database (http://www.ncbi.nlm.nih.gov/sra) (Accession number: PRJNA322600).

**Funding:** This work was supported by the Instituto de Salud Carlos III (ISCII) of the Spanish Ministry of Economy and Competence (MINECO) ISCIII-MINECO AES-FEDER (Plan Estatal de I+D+I 2008-2011 and 2013-2016 https://www.isciii.es/QueHacemos/Financiacion/Paginas/Accion-Estrategica-en-Salud.aspx) (MSB: RD12/0036/0060, PI14/00221, PIE14/0064, PI17/00272; MSB and JGR: CIBERONC CB16/12/00291, MA and MG: CB16/12/00233), Comunidad Autónoma de Madrid (https://www.comunidad.madrid/servicios/educacion/convocatorias-ayudas-investigacion) (MSB: B2017/BMD3778), MSB: Fundación de la Asociación Española Contra el Cáncer (https://www.aecc.es/es/investigacion/fundacion-cientifica-aecc) and JAGM: Roche Farma, S.A., Madrid, Spain. JGR was a recipient of an iPFIS predoctoral fellowship (IFI14/00003) and MSB held a Miguel Servet II contract (CPII16/00024), supported by ISCIII-MINECO AES-FEDER (Plan Estatal I+D+I 2013-2016) and the Fundación de Investigación Biomédica Puerta de Hierro. The funders had no role in study design, data collection and analysis, decision to publish, or preparation of the manuscript.

**Competing interests:** J.A. Garcia-Marco has received honoraria for advisory board and speaker's bureau from Mundipharma, Glaxo, AbbVie, Roche, Gilead, and Janssen, and research support from Hoffman-La Roche and Janssen. Jaime Pérez de Oteyza declares a consulting and advisory role for Hoffman-La Roche. The other authors have no conflicts of interest to declare. This does not alter our adherence to PLOS ONE policies on sharing data and materials.

## Introduction

Chronic Lymphocytic Leukemia (CLL) is the most prevalent leukemia in Western countries and is notable for its variable clinical course. This variability is partly reflected by the mutational status of IGHV genes that defines two subgroups characterized by different clinical outcomes. IGHV-mutated status is associated with long-lasting stable disease and better prognosis, while the IGHV-unmutated genotype (U-IGHV) is associated with a more active and proliferative disease [1–3]. Many CLL samples have been studied in recent years by next-generation sequencing (NGS). These studies have identified recurrent somatic mutations in genes, such as *NOTCH1*, *SF3B1*, *ATM*, *TP53*, *BIRC3*, and others, which play roles in cell cycle, DNA repair, RNA metabolism and splicing, inflammation and NOTCH and WNT signaling pathways[4–6]. Some of them have been found to have prognostic and/or predictive significance. Mutations in *TP53*, *ATM*, *SF3B1* and *NOTCH1* are associated with a significantly shorter time to first treatment and/or overall survival (OS) [4, 7]. In general, patients with a more aggressive disease have higher mutation rates, and patients with shorter progression-free survival (PFS) harbor more mutations per megabase [7]. The standard treatment of choice as first-line therapy for young physically fit patients with CLL is the combination of chemoimmunotherapy (CIT) with fludarabine, cyclophosphamide and rituximab (FCR). Long-term results from three studies [8–10] have demonstrated a long-duration PFS and OS of nearly 12 years in the subset of patients with mutated IGHV and an absence of adverse genetic features (11q deletion [del(11)] or 17p deletion [del(17)]/*TP53* mutation) after treatment with front-line FCR. However, the recent introduction of targeted oral agents, including BTK and BCL2 inhibitors (ibrutinib, acalabrutinib and venetoclax), alone or in combination with monoclonal antibodies (rituximab or obinutuzumab) have demonstrated considerable efficacy in the front-line treatment of patients with CLL with U-IGHV and high-risk cytogenetic biomarkers (del(11q) and del(17p)/*TP53* mutation) [11–13]. However, we do not know the prognostic impact of new recurrent mutations in patients with CLL suitable for front-line immuno-chemotherapy. Indeed, undetectable measurable residual disease (MRD) at the end of treatment is currently the most powerful predictor of clinical outcome related to favorable PFS and prolonged OS in CLL [8, 14].

In this study, we have taken a deep-targeted massive sequencing approach to analyze the impact of mutations in the most frequently mutated genes in a prospectively selected group of patients with CLL with active progressive disease who require treatment. All patients were enrolled in the REM (Rituximab En Mantenimiento [Rituximab in Maintenance]) clinical trial, which consisted of rituximab maintenance for 36 months after achieving at least a partial clinical response to front-line FCR treatment [15].

## Materials and methods

### Patients and samples

Seventy-one peripheral blood samples from treatment-naïve patients with CLL with progressive active disease were included in the present study. The patients were enrolled in the REM clinical trial. REM is a multicenter, non-randomized, prospective phase II clinical trial evaluating the overall response and PFS in patients with CLL with active progressive disease after first-line treatment with FCR, followed by rituximab maintenance every two months for three years in responding patients [15]. Samples were collected at the time of enrollment before treatment. Patient characteristics are summarized in Table 1.

The research project was approved by the Ethics Committee of Hospital Universitario Puerta de Hierro-Majadahonda and conducted following the Declaration of Helsinki. All

**Table 1. Patients' features.**

|  | Category | REM trial |
|---|---|---|
| Gender | Male/Female | 47/24 |
| Age (years) | Median (range) | 59.6 (37–71) |
| Binet stage | A | 12 |
|  | B | 43 |
|  | C | 16 |
|  | 0 | 5 |
| Rai stage | I-II | 47 |
|  | III-IV | 19 |
| Copy number | trisomy 12 | 10/71 (14.1%) |
| alterations | del(13q)/normal | 37/71 (52.1%) |
|  | del(17p) | 3/71 (4.2%) |
|  | del(11q) | 20/71 (28.2%) |
| IGHV | Unmutated | 44/67 (65.4%) |
| CD38 | > 30% positive | 36/69 (52,2%) |
| ZAP70 | > 20% positive | 39/66 (59%) |
| CD49d | > 20% positive | 27/68 (39.7%) |

patients gave their written informed consent for blood collection and the processing of biological analyses included in the present study. The REM study was registered as a clinical trial with NCT#: 00545714 and EudraCT#: 2007-002733-36.

Samples were collected from peripheral blood mononuclear cells (PBMCs) using Ficoll (Rafer, Zaragoza, Spain). Tumor-cell purity was calculated based on the CD19/CD5 ratio, measured by FACS. It ranged from 75% to 98%. DNA was extracted with DNAzol Genomic DNA Isolation Reagent (Molecular Research Center, Cincinnati, OH, USA) following the manufacturer's instructions. The quality and quantity of purified DNA were assessed by fluorimetry (Qubit, Invitrogen, Waltham, MA, USA) and gel electrophoresis.

## Genetic characterization

Cytogenetic aberrations were analyzed by fluorescence *in situ* hybridization (FISH) with the Vysis CLL FISH Probe Kit, following the manufacturer's recommendations for detecting deletions of *TP53* (17p13.1), *ATM* (11q22.3), D13S319 (13q14.3), MYC rearrangements/amplification (8q24.12-q24.13) and gain of the D12Z3 sequence (trisomy 12) in peripheral blood specimens from patients with CLL. Cut-off values for a positive FISH result were 3% and 10% for gains and deletions, respectively.

Amplifications of the IGHV-diversity (D)-joining (J) segment were performed on genomic DNA using standard procedures and analyzed by Sanger sequencing according to ERIC recommendations [16]. IGHV sequences were considered mutated or unmutated using the conventional cut-off of 98% identity with the closest germline IGHV gene.

## Flow cytometry and MRD analysis

Samples were stained and lysed using a direct immunofluorescence technique as previously described [15]. In summary, sequential bone marrow (BM) and peripheral blood (PB) samples were collected in tubes containing K3 EDTA as anticoagulant. BM samples were immediately diluted 1/1 (vol/vol) in phosphate-buffered saline (PBS). Whole BM and PB samples (approximately $2x10^6$ cells in 100 μL per test) were stained and lysed using a direct

immunofluorescence technique, as previously described [15]. The antibody combinations tested were CD22/CD23/CD19/CD5, CD81/CD22/CD19/CD5, CD20/CD38/CD19/CD5, CD20/CD79b/CD19/CD5 and sIgKappa/sIgLambda/CD19/CD5. Cells were acquired in two consecutive steps in order to increase the sensitivity of the analysis. First, 20,000 events corresponding to all nucleated cells were acquired. In the second step, the acquisition was done through a "live gate" drawn on the SSC/CD19+ region in which B-lymphocytes are located. When no CLL cells were detected, to have a limit of detection of 0.01%, a minimum of 20 events was needed and 200,000 events were acquired. To ensure a lower limit of quantification of 0.01%, a minimum of 50 events were required. For ZAP70, CD38 and CD49d measurements see García-Marco *et al*. 2019 [15].

### Targeted massive sequencing

To select genes to be analyzed, we browsed the COSMIC and ICGC databases and reviewed previously published data on CLL (library designed in 2013) [17–26]. Based on their recurrence and prognostic/predictive capacity described in the literature and in our results, the following recurrently mutated genes were selected: *ATM*, *BIRC3*, *BRAF*, *CHD2*, *CSMD3*, *DDX3X*, *FBXW7*, *KLHL6*, *KRAS*, *LRP1B*, *MAPK1*, *MYD88*, *NFKBIE*, *NOTCH1*, *PLEKHG5*, *POT1*, *SAMHD1*, *SF3B1*, *SI*, *SMARCA2*, *TGM7*, *TP53*, *XPO1*, *MUC2*, and *ZMYM3*. *EGR2* (not included in the HaloPlex design) was analyzed independently (see below).

We used two methods for target enrichment and library preparation:

1. A HaloPlex Target Enrichment custom panel was designed using the web-based tool Sure-Design (Agilent Technologies, Santa Clara, CA, USA) (earray.chem.agilent.com/suredesign/). The design covered all coding exons, and UTR regions of the 25 selected genes, including ten flanking bases at the 3'and 5'ends (Human assembly GRCh37/hg19). The final design included 5734 amplicons covering 344,420 bases, with 99.15% of target bases covered by at least one probe. The target regions were captured using a HaloPlex Target Enrichment kit, following the manufacturer's instructions (HaloPlex Target Enrichment System Protocol for Illumina, San Diego, CA, USA). Briefly, 200 ng of genomic DNA was digested with the specific cocktail of restriction enzymes provided in the kit. Digested DNA was then hybridized to a probe for target enrichment, indexed and captured. Each DNA was then amplified by PCR at Tm = 60°C, for 21 cycles, using a Herculase II Fusion Enzyme kit (Agilent Technologies, Santa Clara, CA, USA). Next, amplified target libraries were purified using an Agencourt AMPure XP Kit (Beckman Coulter Genomics, Brea, CA, USA), following the manufacturer's guidelines, and quantified combining Bioanalyzer and Qubit data. Pools were made by combining 5–7 indexed libraries up to a final concentration of 10 nM. Paired-end sequencing was performed in a MiSeq instrument (Illumina) at IMEGEN (Parc Científic de la Universitat de València, Paterna, Valencia, Spain). The mean read depth within the regions of interest was approximately 1181 reads/base (QC data in S1 Table in S1 File)).

2. For hotspots in *EGR2* (a gene not included in the HaloPlex panel), ultra-deep sequencing of specific PCR-based amplification protocol was adopted and sequenced in MiSeq sequencer (Illumina, San Diego, CA, USA). Amplicons of approximately 100 bp were designed with a primer3-based tool (http://bioinfo.ut.ee/primer3-0.4.0/) for the target hotspots in the *EGR2* (S2 Table in S1 File). Independent PCR amplifications were conducted with TaqGold Polymerase (Life Technologies, Carlsbad, CA, USA). Libraries were constructed with the NEB-Next® DNA Library Prep Reagent Set for Illumina (New England Biolabs, Ipswich, MA, USA) following the manufacturer's instructions [27]. A library pool was constructed

combining all the indexed libraries. Paired-end sequencing was performed in a MiSeq instrument. 100% and 66.67% of the analyzable target regions were covered by at least 5,000 and 10,000 reads, respectively (QC data in S1 Table in S1 File).

### Data analysis and variant calling

Data were analyzed using two pipelines: i) MiSeq Reporter alignment, which was performed using the Burrows-Wheeler Aligner (BWA). Variants were identified and annotated with the Genome Analysis Toolkit (GATK); and ii) analysis with SureCall 2.1.13 (Agilent Technologies, Santa Clara, CA, USA) software. The variant lists obtained were analyzed by filtering in Excel and visualizing with the Integrative Genome Viewer (IGV) tool [28].

We applied the filters to identify putative somatic mutations, filtering out those not reaching 100x and those of bad quality (based on the QC Score obtained from MiSeq Reporter). The percentage of reads supporting the mutation from the total number of reads at a given position was taken as 5% in the tumor DNA with a minimum depth of around 200. Only those variants with an allele frequency greater than 20% were considered for validation. Biological impact predictions for detected variants were obtained from the Ensembl Variant Effect Predictor (VEP: http://www.ensembl.org/tools.html), SIFT and PolyPhen predictions for the effect of the mutations on protein function. Variants present in germline DNA or identified as SNPs were excluded from the candidate list.

Matched non-tumoral samples were not available for most patients. The GATK annotates the SNPs available at dbSNP 132 (hg19) and 1000 Genomes Project. Variants present in germline DNA or identified as SNPs were excluded from the candidate list. SNVs with variant allele frequency $\geq$ 5% and not listed as a single nucleotide polymorphism, or listed but with a MAF < 0.01% (The Exome Aggregation Consortium, 1000 Genomes Project of the International Genome Sample Resource (IGSR), Single Nucleotide Polymorphism Database (dbSNP) v132 of the National Center for Biotechnology Information (NCBI)) were considered. Missense, frameshift, and nonsense mutations were selected.

Data have been deposited in the Sequence Read Archive (SRA) database (http://www.ncbi.nlm.nih.gov/sra) (Accession number: PRJNA322600).

### Validation of mutations by Sanger sequencing

A group of selected variants was chosen for validation by Sanger sequencing. According to the following criteria, mutations to be validated were chosen: VAF greater than 20%, not previously described as recurrent, and with sufficient available DNA. Some known variants were also validated. The validation primers (available upon request) were designed with the Primer3 web tool and sequenced with the Big Dye terminator v3.1 Cycle Sequencing kit and an ABI3730 DNA Analyzer (Life Technologies, Carlsbad, CA, USA).

### Statistical analysis

The significance of bivariate relationships between factors was assessed using Pearson's chi-squared or Fisher's exact test; values of $P < 0.05$ were considered significant. Endpoints were PFS, OS and MRD status. OS was calculated from the date of sampling to the date of death or last follow-up, whichever came first. Time to progression was calculated from the date of first treatment to the date of clinical progression or death due to progression. Logistic regression was used to evaluate the association of genetic alterations with MRD. Univariate and multivariate Cox proportional hazard (PH) regression models were used to test the associations of mutations with outcomes. A manual backward selection strategy was used to obtain the final

model, with the criterion for eliminating variables being a significance level of P > 0.05. The PH assumption was tested using Schoenfeld residuals [29]. Hazard ratios, with 95% confidence intervals, were estimated for each parameter. All calculations were performed using IBM SPSS Statistics 19 and STATA v14.1.

## Results and discussion

### Gene mutations and correlation with patients' cytogenetic and phenotypic features

Seventy-one peripheral blood samples from treatment-naïve patients with CLL enrolled in the REM clinical trial [15] with symptomatic, progressive disease were included in our analysis (Table 1). Samples were collected at the time of enrollment in the REM clinical trial (up to 28 days before the first cycle of FCR).

We analyzed the impact of mutations in 26 genes by targeted deep-sequencing as described in the "Methods" section. After sequencing, the median read depth within the regions of interest was 1485 reads/base. A total of 100 mutations were identified in 49/71 (69.0%) patients. Eighteen (25.3%) patients harbored one mutation, whereas 31 (43.6%) had multiple mutations. We did not detect any mutations in 22 patients (31%) (Fig 1 and S3 Table in S1 File).

The most recurrently mutated genes were *SF3B1* (14 mutations in 14 cases; 19.7%) with the recurrent mutation Lys700Glu found in four patients; *NOTCH1* (13 mutations in 12 cases; 16.9% of patients), 8 of them were the previously published indel Pro2515Argfs*4 located in the C-terminal PEST domain of *NOTCH1* (Fig 1 and S2 Table in S1 File); and *ATM* (14 mutations in 9 cases; 12.6%). Other frequently mutated genes were *XPO1* (eight cases; 11.2%), *TP53* (six cases; 8.4%), *CSMD3* (five cases; 7.0%), *EGR2* and *POT1* (in four cases each one; 5.6%) and *LRP1B*, *NFKBIE*, *FBXW7*, *PLEKHG5*, *SAMHD1*, and *SI* (three cases; 4.2%); two out of 3 *NFKBIE* mutations correspond to the previously published indel Tyr245Ile255*16.

Statistical analysis of the presence of gene mutations with CLL phenotypic or cytogenetic characteristics revealed some significant associations (S4 Table in S1 File). The presence of ≥ 2 gene mutations was associated with aggressive CLL features such as U-IGHV (P = 0.017), ZAP70 (P = 0.001) and CD49d expression (P = 0.005). As expected, patients with mutations in *TP53* had a concurrent del(17p) (P < 0.001). Five of the six *TP53*-mutated samples were found in elderly patients (> 65 years; P = 0.001). Eighty-three percent of *NOTCH1*-mutated samples (10/12) showed concurrent mutations in other genes, and *NOTCH1* mutations were associated with the expression of ZAP70 (P = 0.021) CD49d (P = 0.001) and were more frequently found in IGHV-U cases (P = 0.006). It was hypothesized that mutations in *NOTCH1* regulate CD49d expression through the NFkB pathway involvement, favoring drug resistance [30]. All *XPO1*-mutated samples were found in the U-IGHV group (P = 0.044). Finally, mutations in *EGR2* were associated with del(11q) (P = 0.065), and all of them expressed CD38 in more than 30% of CLL cells (although not statistically significant, P = 0.115). By contrast, mutant *LRP1B* was only detected in samples with a normal/del(13q) karyotype (P = 0.048). Therefore, mutations in *NOTCH1* and *XPO1* were enriched among cases with high-risk disease.

### Association with clinical follow-up and response to therapy

Regarding the prognostic significance of gene mutations in patients with CLL with an active progressive disease requiring treatment, we analyzed the clinical significance of these genetic alterations in terms of clinical response, PFS, OS and analyzing their association with measurable MRD (data available for 61 patients) at the end of treatment. Statistical analyses took all the mutations with VAF > 5% into account since we did not find any significant differences

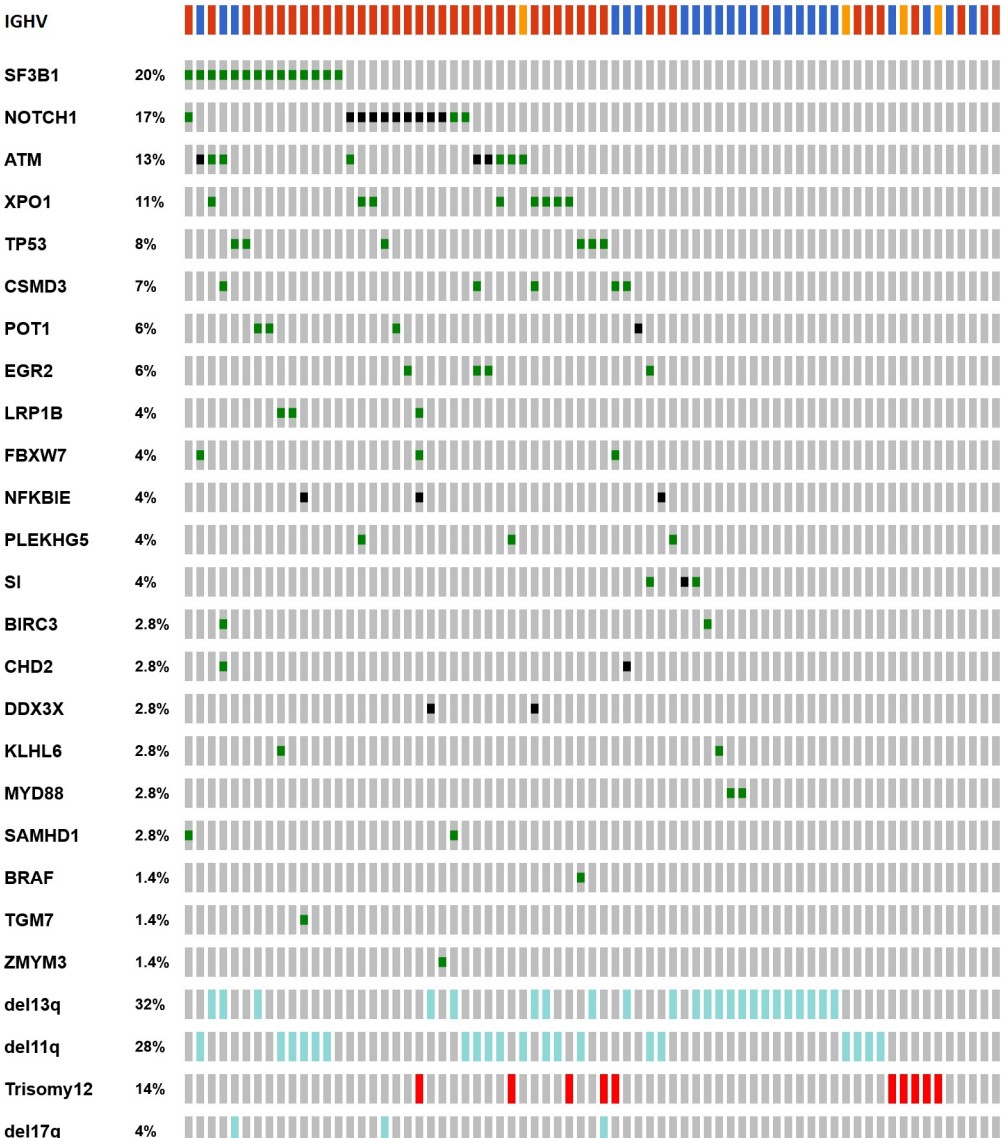

**Fig 1. Incidence and distribution of mutations in the 26 analyzed genes and genetic markers in the REM cohort.**
Rows correspond to sequenced genes and genetic features; columns represent individual patients with CLL.
(Figure done with oncoprint tool in cBioPortal). Colour code: IGHV: red, unmutated IGHV, blue: mutated IGHV,
orange: no data. Genetic alterations: green, missense mutation; black, truncating mutation; light blue, deletion; red,
gain.

between clonal and subclonal mutations. We analyzed del(11q) together with *ATM* mutations
(*ATM*/del(11q)) and del(17p) with *TP53* mutations (*TP53*/del(17p)).

Achieving undetectable MRD remission is the most important predictor of PFS in patients
treated with CIT, independent of clinical remission status and patients' pretreatment charac-
teristics [8, 14], and it is currently accepted by the European Medicines Agency (EMA) as a
surrogate marker for PFS. In our series, undetectable MRD was significantly associated with
prolonged PFS (HR: 6.049, P < 0.001) and so for OS (HR: 3.907, P = 0.044) (S1 Fig in S2 File),
as it is also shown for the whole REM series in the previous publication by García-Marco *et al.*

**Table 2. Univariate and multivariate analysis.** A. Progression-free survival B. Overall Survival.

| | | | | | | | | |
|---|---|---|---|---|---|---|---|---|
| **A** | | | | | | | | |
| | **Univariate** | | | | **Multivariate** | | | |
| | **HR** | **P** | **95.0% CI** | | **HR** | **P** | **95.0% CI** | |
| | | | lower | upper | | | lower | upper |
| **TP53del17p** | 10,234 | <0.001 | 3,936 | 26,609 | 12,843 | <0.001 | 4,724 | 34,920 |
| **EGR2** | 5,052 | 0,014 | 1,394 | 18,310 | 8,256 | 0,002 | 2,114 | 32,238 |
| **IGHV** | 3,572 | 0,019 | 1,237 | 10,310 | | | | |
| **Age < 65y >** | 3,040 | 0,005 | 1,404 | 6,584 | | | | |
| **Binet_2** | 2,719 | 0,012 | 1,245 | 5,937 | | | | |
| **B** | | | | | | | | |
| | **Univariate** | | | | **Multivariate** | | | |
| | **HR** | **P** | **95.0% CI** | | **HR** | **P** | **95.0% CI** | |
| | | | lower | upper | | | lower | upper |
| **TP53del17p** | 10,465 | <0.001 | 3,182 | 34,420 | 10,739 | <0.001 | 3,192 | 36,136 |
| **Binet_2** | 4,094 | 0,009 | 1,426 | 11,749 | 4,164 | 0,009 | 1,437 | 12,242 |
| **Age < 65y >** | 4,054 | 0,010 | 1,396 | 11,726 | | | | |

[15], and in accordance with the criteria established in previous studies [14]. Therefore, we analyzed the correlation between gene mutations and MRD (in 61 patients with MRD data) by logistic regression analysis including cytogenetic abnormalities, IGHV status, number of mutations (0–1 mut vs. ≥ 2 mut) and genes mutated in at least 3 samples (5% of cases): *SF3B1*, *NOTCH1*, *XPO1*, *CSMD3*, *EGR2*, *POT1*, *FBXW7*, *NFKBIE*, and *PLEKHG5*. We analyzed del (11q) together with *ATM* mutations (*ATM*/del(11q)) and del(17p) with *TP53* mutations (*TP53*/del(17p)). Our results showed that additionally to IGHV mutational status (OR: 10,35, P = 0.004), *NOTCH1* mutations (OR: 4,35, P = 0.046), were associated with detectable MDR and, therefore, could be used as predictor of MRD detection together with U-IGHV status.

Finally, we performed univariate Cox PH regressions with each of the following clinical variables: IGHV status, gender, age (> vs. ≤ 65 years), Binet stage (low vs. high risk), cytogenetic alterations and with genes mutated in at least 5% of the patients: *ATM*/del(11q) (24 cases out of 71, 34%), *SF3B1* (14 cases, 20%), *NOTCH1* (12 cases, 17%), *XPO1* (8 cases, 11%), *TP53*/del (17p) (6 cases, 8.5%), *CSMD3* (5 cases, 7%), *EGR2* (4 cases, 5.5%), and *POT1* (4 cases, 5.5%). In the multivariate analyses of the variables that were significant in the univariate analyses (Table 2A and S2A Fig in S2 File), we found that *TP53*/del(17p) (HR: 12.843, P < 0.001) and *EGR2* mutations (HR: 8.256; P = 0.002) increased the risk of progression after treatment. These findings suggest that *EGR2* mutations could be an adverse prognostic biomarker in patients with CLL prospectively treated with FCR followed by R maintenance and could be used as a biomarker to identify patients with poorer outcomes after standard CIT. These results are similar to those reported from the UK LRFCLL4 trial and CLL Research Consortium (CRC) [31], in which alterations in both genes were significantly associated with PFS. The CLL8 study [32] showed that *TP53* and *SF3B1* were the strongest adverse prognostic markers in patients with CLL receiving current-standard first-line therapy; however, *EGR2* was neither associated with PFS nor OS [33].

For OS, additionally to Binet stage only *TP53*mut/del(17p) was found to be an adverse prognostic marker (Table 2B and S2B Fig in S2 File), and also reported by others studies [4, 31, 32].

In conclusion, we have found that the mutation frequencies of several genes by next-generation sequencing, mainly *SF3B1*, *NOTCH1*, and *ATM1*, are similar to those reported in series of patients with CLL requiring therapy.

Also, our results show that the mutational status of patients with CLL in cases that reach an undetectable measurable residual disease at $10^{-4}$ level does not seem to affect the PFS status compared to cases with absence or few gene mutations after front-line FCR treatment followed by limited Rituximab maintenance [15]. Mutations of most recurrent driver genes in CLL, except for the TP53 gene, do not seem to affect the sustained clinical response obtained with front-line FCR treatment followed by Rituximab maintenance for three years in our cohort of patients.

## Supporting information

**S1 File. Supporting tables.** This file contains S1 Table: Coverage and sequencing quality data for HaloPlex and hotspots. S2 Table: Hotspot custom primers for EGR2. S3 Table: Somatic variants from targeted resequencing. S4 Table: Clinical characteristics, cytogenetic abnormalities and molecular markers REM series of 71 patients).
(XLSX)

**S2 File. Supporting figures.** This PDF file contains S1 Fig: MRD association with progression-free survival (A) and overall survival (B). S2 Fig: Progression-free survival (PFS) and overall survival (OS).
(PDF)

## Acknowledgments

We are indebted to the patients who contributed to this study. We thank E. Ramil, A. Royuela and I. Fernández Miranda, from the Instituto de Investigación Sanitaria Puerta de Hierro-Segovia de Arana, and J.L. Bueno from Hospital Universitario Puerta de Hierro-Majadahonda, for their invaluable help with the sequencing (ER) and statistical analysis (AR, IFM and JLB).

## Author Contributions

**Conceptualization:** Margarita Sánchez-Beato, José A. García-Marco.

**Data curation:** José A. Garcia-Vela.

**Formal analysis:** Julia González-Rincón, José A. Garcia-Vela, Miguel Alcoceba, Margarita Sánchez-Beato.

**Funding acquisition:** Marcos González, Margarita Sánchez-Beato, José A. García-Marco.

**Investigation:** Julia González-Rincón, José A. Garcia-Vela, Margarita Sánchez-Beato, José A. García-Marco.

**Methodology:** Julia González-Rincón, Sagrario Gómez, Belén Fernández-Cuevas, Sara Nova-Gurumeta, Nuria Pérez-Sanz.

**Project administration:** Margarita Sánchez-Beato.

**Resources:** Miguel Alcoceba, Marcos González, Eduardo Anguita, Javier López-Jiménez, Eva González-Barca, Lucrecia Yáñez, Ernesto Pérez-Persona, Javier de la Serna, Miguel Fernández-Zarzoso, Guillermo Deben, Francisco J. Peñalver, María C. Fernández, Jaime Pérez de Oteyza, M. Ángeles Andreu, M. Ángeles Ruíz-Guinaldo, Raquel Paz-Arias, M. Dolores García-Malo, Valle Recasens, Rosa Collado, Raúl Córdoba, Belén Navarro-Matilla.

**Supervision:** Julia González-Rincón, Margarita Sánchez-Beato, José A. García-Marco.

**Validation:** Julia González-Rincón, José A. Garcia-Vela, Sagrario Gómez, Eduardo Anguita, Belén Navarro-Matilla, Margarita Sánchez-Beato, José A. García-Marco.

**Visualization:** Julia González-Rincón, Sagrario Gómez, Belén Fernández-Cuevas, Sara Nova-Gurumeta, Nuria Pérez-Sanz, Miguel Alcoceba, Marcos González, Javier López-Jiménez, Eva González-Barca, Lucrecia Yáñez, Ernesto Pérez-Persona, Javier de la Serna, Miguel Fernández-Zarzoso, Guillermo Deben, Francisco J. Peñalver, María C. Fernández, Jaime Pérez de Oteyza, M. Ángeles Andreu, M. Ángeles Ruíz-Guinaldo, Raquel Paz-Arias, M. Dolores García-Malo, Valle Recasens, Rosa Collado, Raúl Córdoba.

**Writing – original draft:** Julia González-Rincón, Margarita Sánchez-Beato, José A. García-Marco.

**Writing – review & editing:** José A. Garcia-Vela, Margarita Sánchez-Beato, José A. García-Marco.

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
