## [Decision Letter · Decision Letter 0]

8 Jun 2021

PONE-D-21-13505

Genomic mutation profile in progressive chronic lymphocytic leukemia patients prior to first-line chemoimmunotherapy with FCR and rituximab maintenance (REM)

PLOS ONE

Dear Dr. Sanchez-Beato,

Thank you for submitting your manuscript to PLOS ONE. After careful consideration, we feel that it has merit but does not fully meet PLOS ONE’s publication criteria as it currently stands. Therefore, we invite you to submit a revised version of the manuscript that addresses the points raised during the review process by the 3 Reviewers, experts in the CLL field.

We look forward to receiving your revised manuscript.

Kind regards,

Francesco Bertolini, MD, PhD

Academic Editor

PLOS ONE

Journal Requirements:

2. Please note that PLOS does not permit references to 'data not shown.' Authors should provide the relevant data within the manuscript, the Supporting Information files, or in a public repository. If the data are not a core part of the research study being presented, we ask that authors remove any references to these data.

4.Thank you for stating the following in the Competing Interests section:

"J.A. Garcia-Marco has received honoraria for advisory board and speaker's bureau from Mundipharma, Glaxo, AbbVie, Roche, Gilead and Janssen, and research support from Hoffman-La Roche and Janssen. Jaime Pérez de Oteyza declares a consulting and advisory role for Hoffman-La Roche. Dr. Gonzalez Barca has received honoraria for speaker's bureau from Roche and AbbVie, and advisory board and speaker's bureau from Janssen, Gilead and Sandoz. The other authors have no conflicts of interest to declare. "

Reviewers' comments:

Reviewer's Responses to Questions

**Comments to the Author**

1. Is the manuscript technically sound, and do the data support the conclusions?

Reviewer #1: Yes

Reviewer #2: Yes

Reviewer #3: Yes

2. Has the statistical analysis been performed appropriately and rigorously? 

Reviewer #1: N/A

Reviewer #2: Yes

Reviewer #3: Yes

3. Have the authors made all data underlying the findings in their manuscript fully available?

Reviewer #1: Yes

Reviewer #2: Yes

Reviewer #3: Yes

4. Is the manuscript presented in an intelligible fashion and written in standard English?

Reviewer #1: Yes

Reviewer #2: Yes

Reviewer #3: Yes

5. Review Comments to the Author

Reviewer #1: The authors sequenced the coding exons and UTR regions of over 20 candidate genes of chronic lymphocytic leukemia (CLL) in 71 CLL patients. They evaluated association between the mutations identified and cytogenetic, phenotypic features and survival rates of the patients.

I have some questions about the data analysis:

Age is a strong player of survival and a dichotomized age variable as a covariate may not be enough to account for age effects. What if age is used as the time scale in Cox regression?

Many of these patients have cytogenetic abnormalities. It is important to evaluate effects of mutations while accounting for the cytogenetic abnormalities, i.e., cytogenetic abnormalities should be included as covariate in the regression analysis.

Lines 302-304: “MRD negativity was significantly associated with prolonged PFS (P Person < 0.001) and nearly so for OS (P Fisher =0.065)” Are the data shown in the result? Why using a Chi-square test in the analysis instead of a survival model?

Logistic regression should be used to evaluate the association with MRD status so that contributions from multiple mutations can be evaluated jointly and covariates such as cytogenetic abnormalities can be adjusted for.

For the Cox regression is the number of mutations per individuals associated with survival? What about the type of genetic alterations (missense, truncating, deletion) of the mutations?

Table 3, was age not significant in the multivariate analysis? As mentioned previously cytogenetic abnormalities needs to be adjusted for.

Line 250, 49/71 seems to be a typo. Shouldn’t the number of patients with at least one mutation be 54 (23+31)?

Reviewer #2: The manuscript by González-Rincón and colleagues reports the results of a deep targeted sequencing approach of the most frequently mutated genes carried out in patients with CLL enrolled in the REM clinical trial where they have been treated with FCR followed by 36 months of rituximab maintenance (bimonthly). They conclude that only TP53 mutations affect the outcome in the study.

I have few suggestions:

- On lines 247-252, where describing the occurrence of mutations it is reported that 69% of patients harbored at least one mutation, while only 23.9% did not. The numbers do not add up to 100. Indeed, it is also reported that 32.4% had one mutations and 43.7% had more than one, for a total of 76.1% which seems more correct. Please double check the numbers or the wording.

- It is interesting to note that SF3B1 is the most frequently mutated gene and much higher than expected in contrast to many other series. How reliable was the VAF in the call for mutations? Any explanation?

- It is not clear which conclusions the authors want to draw. As they state in the abstract "The mutational status of our CLL patients, except for the TP53 gene, does not seem to affect the good results obtained with maintenance therapy with rituximab after front line FCR treatment" though in the last sentence of the paper, they mention a possible use of these analyses to decide between "a continuous versus a time-limited therapy". this does not seem to be a conclusion from this study as they do confirm that the analysis for TP53 status is enough to predict poor response to FCR (despite maintenance). Please revise the wording and reconcile the conclusions.

Minor comments

- Please replace CLL patients with "patients with CLL" throughout the whole text

- Please do not capitalize "del(17p) or del(11q)" not even at the beginning of the sentence or in a table (see e.g. table 1)

- Please on line 135, replace the sentence "....cut-off of a 2% mismatch from germline IGHV sequences." with " cut-off of 98% identity with the closest germline IGHV gene"

- Typically for unmutated IGHV, UM (or U) IGHV is used and not IGHV-U. Consider to align with previous works.

- On line 292, as the authors are planning to explore the predictive value of these gene mutations, the authors might simplify the wording by replacing the sentence "we analyzed the clinical significance of these genetic alterations by studying their effect on the therapeutic response" with "we analyzed the predictive value of these genetic alterations in terms of clinical response, PFS, etc".

- from line 299 onward, please replace "MRD negative/negativity" with "undetectable MRD" or "detectable MRD" instead of "positivity".

Reviewer #3: The manuscript reports on an ancillary biological study of a phase 2 trial where patients with treatment naive CLL received FCR, followed by rituximab maintenance every two months for three years. The study includes 71 patients, whose pre-treatment leukemia samples were profiled for CLL biomarkers including IGHV and targeted gene mutations, and cytogenetics. The study cohort is also provided by longitudinal MRD data. The results further validate previous findings that TP53 abnormalities are the sole lesions associated with outcome. Accordingly, in its current form, the manuscript does not add any further novel finding compared to what already known in the field of prognostic biomarkers of CLL treated with chemoimmunotherapy. In addition, from a conceptual standpoint, FCR has been largely abandoned thanks to the transition to pathway inhibitor therapies in CLL. Thus the results are of eventual historical interest, but do not have actual implications. Finally maintenance with rituximab is not a standard approach in CLL and the study does not help in signaling those patients whose molecular profile may benefit or not from maintenance. Novelty of the manuscript can be substantially improved in the authors can validate or further develop upon AI methods, that leveraging on big data, frameworks for the integration of patient biomarker data over time to improve prognostic accuracy and personalized therapy selection (PMID: 31280963).

6. PLOS authors have the option to publish the peer review history of their article (what does this mean?). If published, this will include your full peer review and any attached files.

Reviewer #1: No

Reviewer #2: No

Reviewer #3: No

---

## [Author Response · Author response to Decision Letter 0]

7 Jul 2021

Reviewer #1: 

The authors sequenced the coding exons and UTR regions of over 20 candidate genes of chronic lymphocytic leukemia (CLL) in 71 CLL patients. They evaluated association between the mutations identified and cytogenetic, phenotypic features and survival rates of the patients.

I have some questions about the data analysis:

Age is a strong player of survival and a dichotomized age variable as a covariate may not be enough to account for age effects. What if age is used as the time scale in Cox regression? 

R: We considered including the variable “Age” as a continuous variable in the Cox regression analysis since we agree with the reviewer that it is more accurate. However, we eventually decided to dichotomize this variable to use it in the multivariate analysis with other dichotomic variables and ease the comparison with other studies, where “Age” is used as a categoric variable. The threshold at 65 years is the usual standard in most multivariate analysis published in recent years. However, we did the analysis using age as a continuous variable but did not reach significance (P = 0.072).

Many of these patients have cytogenetic abnormalities. It is important to evaluate effects of mutations while accounting for the cytogenetic abnormalities, i.e., cytogenetic abnormalities should be included as covariate in the regression analysis.

R: Cytogenetic abnormalities were included in the analyses, but none were significant in the regression analysis except when TP53 mutations and del17q were joined as a single variable.

Lines 302-304: “MRD negativity was significantly associated with prolonged PFS (P Person < 0.001) and nearly so for OS (P Fisher =0.065)” Are the data shown in the result? Why using a Chi-square test in the analysis instead of a survival model? 

R: Thank you for the suggestion. We have done the survival models (Cox and Kaplan-Meier) and included them in the manuscript (Page 14, line 306, and Figure S1). 

Logistic regression should be used to evaluate the association with MRD status so that contributions from multiple mutations can be evaluated jointly and covariates such as cytogenetic abnormalities can be adjusted for. 

R: Thank you for the suggestion. We have used logistic regression and included the results in page 14, line 310.

For the Cox regression is the number of mutations per individuals associated with survival? What about the type of genetic alterations (missense, truncating, deletion) of the mutations? 

R: Yes, we considered the number of mutations in the survival analysis, but there was no association with OS or PFS. Due to the low number of samples, we considered that we would not obtain relevant results by splitting the type of mutations. The appropriate analysis would be done for each gene independently, and our series does not allow us to do this study properly.

Table 3, was age not significant in the multivariate analysis? As mentioned previously cytogenetic abnormalities needs to be adjusted for. 

R: Age was significant in the univariate analysis, as indicated in now table 2, but loses significance in the multivariate analyses. None of the cytogenetic alterations was significant, except for TP53/del17q, as shown in the table.

Line 250, 49/71 seems to be a typo. Shouldn’t the number of patients with at least one mutation be 54 (23+31)?

R: Thank you for the observation. We have corrected this and other errors.

Reviewer #2: 

The manuscript by González-Rincón and colleagues reports the results of a deep targeted sequencing approach of the most frequently mutated genes carried out in patients with CLL enrolled in the REM clinical trial where they have been treated with FCR followed by 36 months of rituximab maintenance (bimonthly). They conclude that only TP53 mutations affect the outcome in the study.

I have few suggestions:

- On lines 247-252, where describing the occurrence of mutations it is reported that 69% of patients harbored at least one mutation, while only 23.9% did not. The numbers do not add up to 100. Indeed, it is also reported that 32.4% had one mutations and 43.7% had more than one, for a total of 76.1% which seems more correct. Please double check the numbers or the wording. 

R: We apologize for these mistakes; the numbers have been revised and corrected.

- It is interesting to note that SF3B1 is the most frequently mutated gene and much higher than expected in contrast to many other series. How reliable was the VAF in the call for mutations? Any explanation? 

R: According to our experience, data in previous publications, and sequencing depth, we think that VAF is reliable, as we described in the Methods section (page 9, line 200).

One possible explanation might be the characteristics of the series, composed by progressive cases in need of treatment compared to others that considered any kind of patient (see page 16, line 347).

- It is not clear which conclusions the authors want to draw. As they state in the abstract “The mutational status of our CLL patients, except for the TP53 gene, does not seem to affect the good results obtained with maintenance therapy with rituximab after front line FCR treatment” though in the last sentence of the paper, they mention a possible use of these analyses to decide between “a continuous versus a time-limited therapy”. This does not seem to be a conclusion from this study as they do confirm that the analysis for TP53 status is enough to predict poor response to FCR (despite maintenance). Please revise the wording and reconcile the conclusions.

R: Thank you for the suggestion. We have changed the last sentence of the discussion. We hope it would help to clarify our conclusion.

“Mutations of most recurrent driver genes in CLL, except for the TP53 gene, do not seem to affect the sustained clinical response obtained with front-line FCR treatment followed by Rituximab maintenance for three years in our cohort of patients”.

Minor comments

- Please replace CLL patients with “patients with CLL” throughout the whole text

- Please do not capitalize “del(17p) or del(11q)” not even at the beginning of the sentence or in a table (see e.g. table 1)

- Please on line 135, replace the sentence “....cut-off of a 2% mismatch from germline IGHV sequences.” with “cut-off of 98% identity with the closest germline IGHV gene”

- Typically for unmutated IGHV, UM (or U) IGHV is used and not IGHV-U. Consider to align with previous works. 

- On line 292, as the authors are planning to explore the predictive value of these gene mutations, the authors might simplify the wording by replacing the sentence “we analyzed the clinical significance of these genetic alterations by studying their effect on the therapeutic response” with “we analyzed the predictive value of these genetic alterations in terms of clinical response, PFS, etc”.

- from line 299 onward, please replace “MRD negative/negativity” with “undetectable MRD” or “detectable MRD” instead of “positivity”.

R: Thank you for the suggestions. We have modified them in this new version and corrected this and other mistakes.

Reviewer #3: 

The manuscript reports on an ancillary biological study of a phase 2 trial where patients with treatment naive CLL received FCR, followed by rituximab maintenance every two months for three years. The study includes 71 patients, whose pre-treatment leukemia samples were profiled for CLL biomarkers including IGHV and targeted gene mutations, and cytogenetics. The study cohort is also provided by longitudinal MRD data. The results further validate previous findings that TP53 abnormalities are the sole lesions associated with outcome. Accordingly, in its current form, the manuscript does not add any further novel finding compared to what already known in the field of prognostic biomarkers of CLL treated with chemoimmunotherapy. In addition, from a conceptual standpoint, FCR has been largely abandoned thanks to the transition to pathway inhibitor therapies in CLL. Thus the results are of eventual historical interest, but do not have actual implications. Finally maintenance with rituximab is not a standard approach in CLL and the study does not help in signaling those patients whose molecular profile may benefit or not from maintenance. Novelty of the manuscript can be substantially improved in the authors can validate or further develop upon AI methods, that leveraging on big data, frameworks for the integration of patient biomarker data over time to improve prognostic accuracy and personalized therapy selection (PMID: 31280963).

R: A weakness of this trial is that over the last few years, since its design, new drugs targeting aberrant signaling pathways and newer monoclonal antibodies became available, and the interest in CIT, such as FCR, has weakened. However, it is still accepted as front-line therapy by some clinical guidelines (German, Spanish-CLL, ESMO, etc.) in a subgroup of patients younger than 65 years without comorbidities and absence of cytogenetic abnormalities of worse prognosis (del11q, del17p, complex karyotypes or unmutated IGHV).

Unfortunately, REM clinical trial series has not enough data to analyze them using complex algorithms. Moreover, we only have one-point data (from treatment-naïve samples), not serial samples, precluding us from using analyses like those performed in the suggested paper that integrated risk assessments throughout patients’ disease evolution.

---

## [Decision Letter · Decision Letter 1]

28 Jul 2021

PONE-D-21-13505R1

Genomic mutation profile in progressive chronic lymphocytic leukemia patients prior to first-line chemoimmunotherapy with FCR and rituximab maintenance (REM)

PLOS ONE

Dear Dr. Sanchez-Beato,

Thank you for submitting your manuscript to PLOS ONE. After careful consideration, we feel that it has merit but does not fully meet PLOS ONE’s publication criteria as it currently stands. Therefore, we invite you to submit a revised version of the manuscript that addresses the points raised during the review process by Reviewer #1.

We look forward to receiving your revised manuscript.

Kind regards,

Francesco Bertolini, MD, PhD

Academic Editor

PLOS ONE

Journal Requirements:

Reviewers' comments:

Reviewer's Responses to Questions

**Comments to the Author**

1. If the authors have adequately addressed your comments raised in a previous round of review and you feel that this manuscript is now acceptable for publication, you may indicate that here to bypass the “Comments to the Author” section, enter your conflict of interest statement in the “Confidential to Editor” section, and submit your "Accept" recommendation.

Reviewer #1: All comments have been addressed

Reviewer #3: All comments have been addressed

2. Is the manuscript technically sound, and do the data support the conclusions?

Reviewer #1: Yes

Reviewer #3: Yes

3. Has the statistical analysis been performed appropriately and rigorously? 

Reviewer #1: Yes

Reviewer #3: Yes

4. Have the authors made all data underlying the findings in their manuscript fully available?

Reviewer #1: (No Response)

Reviewer #3: Yes

5. Is the manuscript presented in an intelligible fashion and written in standard English?

Reviewer #1: Yes

Reviewer #3: Yes

6. Review Comments to the Author

Reviewer #1: I only have a minor comment, please specify what variables/covariates were included in the logistic regression.

Reviewer #3: The authors have addressed all the reviewer comments and the manuscript is improved in this current version. I have no further issues.

7. PLOS authors have the option to publish the peer review history of their article (what does this mean?). If published, this will include your full peer review and any attached files.

Reviewer #1: No

Reviewer #3: No

---

## [Author Response · Author response to Decision Letter 1]

5 Aug 2021

Reviewer 1: I only have a minor comment, please specify what variables/covariates were included in the logistic regression.

Thank you for drawing to our attention this ommision. We have included a sentence indicating the varibles included in the logistic regression analyis (Page 14, lines 312-316).

---

## [Decision Letter · Decision Letter 2]

31 Aug 2021

Genomic mutation profile in progressive chronic lymphocytic leukemia patients prior to first-line chemoimmunotherapy with FCR and rituximab maintenance (REM)

PONE-D-21-13505R2

Dear Dr. Sanchez-Beato,

We’re pleased to inform you that your manuscript has been judged scientifically suitable for publication and will be formally accepted for publication once it meets all outstanding technical requirements.

Kind regards,

Francesco Bertolini, MD, PhD

Academic Editor

PLOS ONE

Additional Editor Comments (optional):

Reviewers' comments:

Reviewer's Responses to Questions

**Comments to the Author**

1. If the authors have adequately addressed your comments raised in a previous round of review and you feel that this manuscript is now acceptable for publication, you may indicate that here to bypass the “Comments to the Author” section, enter your conflict of interest statement in the “Confidential to Editor” section, and submit your "Accept" recommendation.

Reviewer #1: All comments have been addressed

2. Is the manuscript technically sound, and do the data support the conclusions?

Reviewer #1: (No Response)

3. Has the statistical analysis been performed appropriately and rigorously? 

Reviewer #1: (No Response)

4. Have the authors made all data underlying the findings in their manuscript fully available?

Reviewer #1: (No Response)

5. Is the manuscript presented in an intelligible fashion and written in standard English?

Reviewer #1: (No Response)

6. Review Comments to the Author

Reviewer #1: (No Response)

7. PLOS authors have the option to publish the peer review history of their article (what does this mean?). If published, this will include your full peer review and any attached files.

Reviewer #1: No

---

## [Editor Report · Acceptance letter]

3 Sep 2021

PONE-D-21-13505R2 

Genomic mutation profile in progressive chronic lymphocytic leukemia patients prior to first-line chemoimmunotherapy with FCR and rituximab maintenance (REM) 

Dear Dr. Sánchez-Beato:

I'm pleased to inform you that your manuscript has been deemed suitable for publication in PLOS ONE. Congratulations! Your manuscript is now with our production department. 

Kind regards, 

on behalf of

Dr. Francesco Bertolini 

Academic Editor

PLOS ONE